# Chemoattraction of Neoplastic Glial Cells with CXCL10, CCL2 and CCL11 as a Paradigm for a Promising Therapeutic Approach for Primary Brain Tumors

**DOI:** 10.3390/ijms222212150

**Published:** 2021-11-10

**Authors:** Laurence Déry, Gabriel Charest, Brigitte Guérin, Mohsen Akbari, David Fortin

**Affiliations:** 1Department of Nuclear Medicine and Radiobiology, Université de Sherbrooke, Sherbrooke, QC J1H 5N4, Canada; Brigitte.Guerin2@USherbrooke.ca; 2Department of Surgery, Division of Neurosurgery, Centre Hospitalier Universitaire de Sherbrooke, Sherbrooke, QC J1H 5N4, Canada; Gabriel.Charest@USherbrooke.ca (G.C.); David.Fortin@USherbrooke.ca (D.F.); 3Laboratory for Innovation in Microengineering (LiME), Department of Mechanical Engineering, University of Victoria, Victoria, BC V8P 5C2, Canada; makbari@uvic.ca; 4Biotechnology Center, Silesian University of Technology, Akademicka 2A, 44-100 Gliwice, Poland

**Keywords:** neuro-oncology, brain tumor, glioblastoma, chemokine, migration

## Abstract

Chemoattraction is a normal and essential process, but it can also be involved in tumorigenesis. This phenomenon plays a key role in glioblastoma (GBM). The GBM tumor cells are extremely difficult to eradicate, due to their strong capacity to migrate into the brain parenchyma. Consequently, a complete resection of the tumor is rarely a possibility, and recurrence is inevitable. To overcome this problem, we proposed to exploit this behavior by using three chemoattractants: CXCL10, CCL2 and CCL11, released by a biodegradable hydrogel (GlioGel) to produce a migration of tumor cells toward a therapeutic trap. To investigate this hypothesis, the agarose drop assay was used to test the chemoattraction capacity of these three chemokines on murine F98 and human U87MG cell lines. We then studied the potency of this approach in vivo in the well-established syngeneic F98-Fischer glioma-bearing rat model using GlioGel containing different mixtures of the chemoattractants. In vitro assays resulted in an invasive cell rate 2-fold higher when chemokines were present in the environment. In vivo experiments demonstrated the capacity of these specific chemoattractants to strongly attract neoplastic glioblastoma cells. The use of this strong locomotion ability to our end is a promising avenue in the establishment of a new therapeutic approach in the treatment of primary brain tumors.

## 1. Introduction

Despite several major advances in recent years, and the constant evolution of technologies in neuroscience, there has been very little progress in the management of patients with glioblastoma (GBM). This grade IV primary brain tumor represents about 50% of malignant neoplasms of the central nervous system (CNS) [1,2,3]. The origin of this neoplasm is not yet determined, but many phenotypic and molecular characteristics are extremely similar to astrocytes and/or cancer stem cells (CSCs) [4,5]. Astrocytes are normally involved in several essential functions such as neuronal repair, absorption and release of neurotransmitters and maintenance of the integrity of the blood–brain barrier (BBB) [6]. The standard treatment has been unchanged for over 15 years and provides a median survival of only 14.6 months. It consists of maximum resection of the tumor followed by concomitant adjuvant radio-chemotherapy treatments with temozolomide [7]. This multimodal treatment, although fairly aggressive, offers only limited effectiveness and a very diverse clinical response [8]. The limited success in the treatment of this affliction derives from the heterogeneous nature of the disease, the presence of the BBB, as well as the very high migration capacity of tumor cells in the cerebral parenchyma, hence rendering it impossible to eradicate all neoplastic cells [9,10]. Therefore, tumor recurrence is inevitable and despite the efforts of the scientific community to establish a second-line treatment, no standard of care has yet been emerged [11,12].

To improve the outcome in the treatment of malignant gliomas, several research groups have devoted efforts in understanding how GBM tumor cells manage to infiltrate brain tissue, and how this mechanism can be halted. Henceforth, it has been clearly demonstrated that the interaction of glial cancer cells with the microenvironment promotes tumor heterogeneity and increases the infiltration of neoplastic cells [13,14]. According to some studies, reactive astrocytes that are mainly found at the periphery of the tumor could stimulate the epithelial–mesenchymal transition (EMT) and favor a migratory phenotype [15,16]. In addition, these astrocytes are a major source of expression of cytokines involved in cell migration and recruitment of various factors [6,17]. These small signaling molecules, known as chemokines or chemoattractants, play a very important role in the development of the immune response [18]. They induce chemotaxis, by specifically binding receptors coupled to G proteins expressed at the surface of target cells [19]. Interestingly, it has been observed that some patients with GBM overexpress various chemokines and their receptors, such as CXCL10 (CXCR3) [20], CCL2 (CCR2) [21] and CCL11 (CCR3) [22]. This overexpression is indeed strongly correlated with an increase in the migration and invasion of tumor cells in the brain parenchyma [23,24]. Unsuccessful therapeutic strategies have been attempted to block this migrational phenotype of the tumor cells, such as the use of the drug cilengitide, a cyclized RGD pentapeptide ανβ3 and ανβ5 integrin inhibitor [25].

What if, instead of blocking this migratory potential, we would use it in the development of a new therapeutic approach? Herein, we hypothesized that it might be possible to attract tumor cells scattered around a surgical resection cavity into a trap using one or several chemokines delivered in the local environment. As the main caveat of a broader composite strategy, we have therefore developed a biodegradable device consisting of hydrogel (GlioGel), implantable directly at the surgical resection site, and allowing local release of different classes of agents [26]. This hydrogel device would contain chemoattractants, serving the purpose of luring infiltrated glial tumor cells toward the resection cavity containing the GlioGel for intensive regional treatment. In the present study, we surveyed and demonstrated the potency of the chemokines CXCL10, CCL2 and CCL11 to attract neoplastic glioblastoma cells in vitro and in vivo. In addition, various chemo-radiotherapeutic agents can be delivered locally to eliminate the scattered tumor cells, hence reducing the side effects linked to systemic toxicity. This design would also allow to bypass the blood–brain barrier, hence circumventing delivery impediment of therapeutics to the CNS and could potentially significantly increase the therapeutic effect, while limiting systemic side-effects. This approach shows new possibilities of personalized treatments for primary brain tumors, but also for all other diseases where complete resection is difficult to achieve.

## 2. Results

### 2.1. In Vitro Chemoattraction Assays

The first experiment evaluated the chemoattraction potential of the chemoattractants CXCL10, CCL2, CCL11 individually and in combination on two established glioblastoma cell lines: F98 and U87MG. Hence, to do so, we carried out the in vitro chemoattraction using the agarose drop assay, as previously described by Ahmed’s group [27]. All experiments and respective analysis were conducted with a test drop (containing each chemokine, or the combination of all three) and with a control drop containing phosphate buffer saline (PBS) (without chemokine), in triplicate for each group. This assay allows to assess the migration of the tumor cells triggered by the gradient of chemokine in the medium. In summary, CXCL10 elicited the greatest effect in both cell types. Figure 1 illustrates the methodology used and presents the detailed results for CXCL10. Indeed, the statistical difference of the cell count per zone comparing control condition to CXCL10 appears fairly robust. The results of the mean values obtained for individual chemokine (CXCL10, CCL2, and CCL11) and the combination are presented in Figure 2.

As can be appreciated in Figure 1A, the number of F98 cells is significantly higher in the presence of the CXCL10 than under the control condition (orange box). Figure 1B presents the quantitative analysis and shows the values for each triplicate (*upper left*) and the average number of cells per zone (*upper right*). The zones where the chemokine concentration is highest (zone 0 and zone 1) proportionally display the highest number of cells compared to the control condition. Interestingly, we also noticed that this effect seems to diminish in zone 2 and 3, as we moved away from the drop, in function of the presumed chemoattractant gradient. The average number of cells per zone for F98 was as follows for control and CXCL10, respectively, **zone 0**: 397 vs. 1873 cells/mm^2^ (*p* = 0.0007), **zone 1**: 781 vs. 2087 cells/mm^2^ (*p* = 0.0018), **zone 2**: 355 vs. 744 cells/mm^2^ (*p* = 0.3944) and **zone 3**: 591 vs. 768 cells/mm^2^ (*p* = 0.5775). The same phenomenon was observed with the U87MG cells for control and CXCL10 respectively, **zone 0**: 573 vs. 1170 cells/mm^2^ (*p* = 0.0039), **zone 1**: 577 vs. 1165 cells/mm^2^ (*p* = 0.0039), **zone 2**: 477 vs. 811 cells/mm^2^ (*p* = 0.0769) and **zone 3**: 673 vs. 873 cells/mm^2^ (*p* = 0.1974).

The second experiment consisted in the evaluation of the maximum migration distance inside the drop for the 3 most invasive cells. As can be seen in Figure 1A for the F98 cells, the migration distance inside the drop is significantly higher in the presence of CXCL10 than under control condition (double heads yellow arrows). Figure 1B presents the quantitative analysis and shows the values for each replicate (lower left) and the average migration distance for each triplicate (lower right). The average values of migration distance for control and CXCL10 respectively is: 0.229 vs. 0.421 mm for triplicate 1 (*p* = 0.02), 0.234 vs. 0.596 mm for triplicate 2 (*p* = 0.0002) and 0.236 vs. 375 mm for triplicate 3 (*p* = 0.04). Interestingly, we observed similar behavior for the U87MG cells in Figure 1C, but at a lower level. The average values of migration distance are detailed in Figure 1D: 0.121 vs. 0.161 mm for triplicate 1 (ns), 0.147 vs. 0.217 mm for triplicate 2 (*p* = 0.05) and 0.152 vs. 0.231 mm for triplicate 3 (*p* = 0.04) for control and CXCL10, respectively.

As can be seen in Figure 2, CCL2 produced a very similar chemotactic response than CXCL10 for the F98 cell line, with control and CCL2 count of 630 vs. 1394 cells/mm^2^ (*p* = 0.0025) respectively for **zone 0**, and of 928 vs. 2010 cells/mm^2^ (*p* = 0.0001) for **zone 1**. The difference in chemoattraction potential of U87MG appears slightly less significant, with count values of 445 vs. 949 cells/mm^2^ (*p* = 0.0061) for control and CCL2 respectively in **zone 0** and 387 vs. 676 cells/mm^2^ (*p* = 0.1272) in **zone 1.**

In contrast to these findings, CCL11 did not demonstrate any significant effect on chemotaxis, neither in F98 nor in U87MG tumor cells. As for the combination of the three chemoattractant, the trends observed with CXCL10 and CCL2 were maintained but appeared less impactful. F98 cell counts were as follows for control and combination respectively, in **zone 0**: 929 vs. 1589 cells/mm^2^ (*p* = 0.0158) and **zone 1**: 1212 vs. 2130 cells/mm^2^ (*p* = 0.0015) whereas the following counts for U87MG cells for control and combination, in **zone 0**: 337 vs. 733 cells/mm^2^ (*p* = 0.0464), and **zone 1**: 278 vs. 474 cells/mm^2^ (*p* = 0.4478) were found. Additional data reported in Appendix A also present the detailed chemoattraction potential results for each chemoattractant.

The mean values are reported for all chemoattractants, as well as for the combination in Figure 2B,D for F98 and U87MG cell lines respectively. In both cell types, the cell migration distance in control condition was significantly lower compared to those obtained in the presence of CXCL10: 0.233 vs. 0.464 mm (*p* = 0.0001) for F98 cells and 0.140 vs. 0.202 mm (*p* = 0.0032) for U87MG cell. Very similar results were found for CCL2, with a mean distance for control and CCL2 respectively, of 0.236 vs. 0.424 mm (*p* = 0.0001) for F98 cells and 0.183 vs. 0.312 mm (*p* = 0.0001) for U87MG cells. Here again, CCL11 was the least impactful chemokine, with 0.214 vs. 0.266 mm (*p* = 0.0579) for F98 cells and 0.130 vs. 0.207 mm (*p* = 0.0397) for U87MG cells, both for control and CCL11 respectively. A comparable value result to CXCL10 was obtained for the combination, but at a lower level of significance, with distances of 0.250 vs. 0.321 mm (*p* = 0.0068) for F98 cells and 0.254 vs. 0.429 mm (*p* = 0.0018) for U87MG cells, both for control and combination respectively. Additional data of migration distance results are presented in Appendix A for each chemoattractant.

### 2.2. In Vivo Chemotactic Response to the Inoculation of GlioGel Containing Chemokine

The chemoattraction potency of the chemokine was then investigated in vivo using the orthotopic syngeneic F98-Fischer rat model. On day 3 post implantation of F98 tumor cells, inoculation of the GlioGel with (test condition) or without (control condition) chemoattractant was performed. Two sets of experiments were realized with the aim of painting the most adequate picture as to the potency of deploying this approach in vivo. The GlioGel was inoculated at the same coordinates as the tumor cells implantation either (1) intratumorally (ipsilateral), or (2) in the mirroring coordinates of the controlateral hemisphere (Figure 3A). The goals of both experiments were obviously divergent: the intratumoral (ipsilateral) inoculation was performed with the aim of assessing the potency of the GlioGel with chemoattractants in limiting tumor cells migration in the brain parenchyma. Conversely, the inoculation of GlioGel containing chemoattractants in the controlateral hemisphere was performed to assess the potency in increasing tumor migration away from the implantation site, as in luring tumor cells toward the controlateral implanted hemisphere. The brains were harvested on day 15 post tumor implantation, twelve days after the GlioGel inoculation.

A total of 88 animals were accrued in these experiments and 8 animals per group were initially planned. However, 11 specimens could not be included in the final data analysis: 4 histology slides did not show visible tumor, 2 had dissociated cell clumps at the site of tumor cell implantation and 5 were altered with the subsequent histological procedures leaving a total of 77 animals for analysis. The Weka segmentation plugin of the Fiji software was used to visualize the presence of peritumoral clusters on hematoxylin and eosin coloration (H&E) (Figure 3B). Peritumoral cluster count was used to describe the impact of the chemoattractant on tumor cells (Figure 3C–E). Interestingly and as expected, the inoculation of the GlioGel containing chemokines in both experiments modified the pattern of tumor growth compared to control conditions (GlioGel with PBS). The detailed data for each category and for each comparison are summarized in Appendix A.

#### 2.2.1. Ipsilateral GlioGel Inoculation

The inoculation of GlioGel containing chemokines within the confines of the tumor nodule appeared to slow the spread of tumor cell dissemination in the surrounding parenchyma. This can be appreciated by a tumor nodule that appears well delimited with an important decrease in the number of cellular clusters at the periphery of the tumor, compared to control conditions (Figure 3A, right). Hence, when looking specifically at peritumoral tumor cell clusters as a surrogate of glial migration and infiltration, the average results between control condition (8 animals) and all chemokine groups were highly significant: **CXCL10** (7 animals): 37 vs. 7 (*p* < 0.0001), **CCL2** (9 animals): 37 vs. 28 (*p* = 0.0258), **CCL11** (8 animals): 37 vs. 25 (*p* = 0.0003) and the **combination** (7 animals): 37 vs. 24 (*p* = 0.0070) (Figure 3D).

#### 2.2.2. Controlateral GlioGel Inoculation

When the GlioGel with chemokines were inoculated in the controlateral hemisphere (Figure 3C), we observed an increase in tumor cell dissemination with several cellular clusters at the periphery of the tumor implantation site. In addition, a satellite cluster in the controlateral hemisphere close to the GlioGel inoculation site is visible in one sample (Figure 3A, left). Hence, comparing average peritumoral cluster counts of control condition (8 animals) to that for GlioGel contain chemokines lend to statistical differences in all but one group (combination); **CXCL10** (7 animals): 44 vs. 65 (*p* = 0.0269), **CCL2** (8 animals): 44 vs. 84 (*p* < 0.0001), **CCL11** (8 animals): 44 vs. 61 (*p* = 0.0191) and the **combination** (7 animals): 44 vs. 53 (*p* = 0.1709) (Figure 3C). It would therefore seem that chemokines tend to increase peritumoral clusters, as if able to increase the potency of migration in this animal model, when implanted at a distance from the tumor nodule.

#### 2.2.3. Ipsilateral GlioGel Inoculation Compared to Controlateral GlioGel Inoculation

As anticipated, when we compared the number of peritumoral clusters in the presence of the different chemokines between each experimental arm, the controlateral vs. intratumoral inoculation (Figure 3E), we observed a significantly higher number of peritumoral clusters in the controlateral GlioGel inoculation groups, **CXCL10**: 65 vs. 7 (*p* = 0.0002), **CCL2**: 84 vs. 28 (*p* < 0.0001), **CCL11**: 61 vs. 25 (*p* = 0.0003) and the **combination**: 53 vs. 24 (*p* = 0.0014). This observation is in keeping with the presumed effect of the chemokines on the migration of glial tumor cells. Interestingly, control conditions in both the controlateral and intratumoral inoculation groups displayed a similar number of clusters 44 vs. 37 (*p* > 0.05, ns). This suggests the adequacy of the control conditions and the significant intended effect of chemokines on the migration of glial tumor cells. It indeed appears that GlioGel with chemokines inoculated in the tumor limits tumor cells spread, whereas the controlateral inoculation presents an increase in migration.

### 2.3. Immunohistochemistry (IHC) of GlioGel Inoculation

To better characterize the impact of the selected chemokine, use on the migrating tumor cells in vivo, we undertook a panel of immunohistochemical studies and a quantification analysis of one sample from each of the 10 groups. We elected to label samples with GFAP, nestin and Ki-67 as well as CD86 and CD68 for macrophages and a granulocyte marker. Our neuropathologist colleague carried out the cytological analysis and confirmed that there was no or negligible presence of lymphocytes. In summary, the inoculation of GlioGel with chemoattractants increases the labelling of all markers compared to control condition and this tendency was observed in all treated groups (Figure 4). When the GlioGel is inoculated in the tumor, most of the marking for GFAP was at the periphery of the tumor. In the controlateral hemisphere, the marker also stains the area around the GlioGel insertion site (Figure 4, left). For nestin labelling, when GlioGel with chemokine is inoculated in the tumor, we observed an increase compared to the control GlioGel. The same phenomenon was observed when the GlioGel with chemokine is implanted in the controlateral hemisphere (Figure 4, center). In the ipsilateral hemisphere, Ki-67 marking is predominantly intratumoral. Conversely, in the controlateral hemisphere, Ki-67 labeling was mainly at the periphery of the GlioGel inoculation site (Figure 4, right).

When attempting quantification of staining, we found that the lowest staining score for all markers is found in the left hemisphere (Figure 5A), in the situation where there is no tumor and no GlioGel. On the other hand, the right hemisphere bearing the tumor with (Figure 5B) and without GlioGel inoculation (Figure 5D), shows a strong expression of all markers. This effect was increased by the presence of chemokines, whether ipsi- or controlateral, by up to a 4-fold factor (Figure 5). In addition, the quantitative analysis of nestin reports a pronounced increase in the tumor bearing hemisphere compared to the baseline staining, suggesting the presence of stem cells within the tumor (Figure 5B vs. Figure 5A and Figure 5D vs. Figure 5C, Ctrl condition). The Ki-67 marking was increased in the presence of chemoattractants, whether ipsi- or controlateral inoculation. The inflammation markers showed extremely scarce staining: CD86 did not present significant staining, and neither did CD68 but depicted unusual marking mostly at the superficial periphery of the tumor (Appendix A). The marker for granulocytes depicted no observable staining.

One potential pitfall of our approach was that the use of chemokines might also modify phenotypic characteristics, notably proliferation. We compared the fraction staining scores for Ki-67 with the peritumoral clusters (Figure 5, numerical value in the IHC image). Interestingly, the inoculation of GlioGel with chemokines increased Ki-67 inside the tumor nodule, but the number of peritumoral clusters remained significantly lower than the control.

## 3. Discussion

Chemoattractants appear to be involved in the migration and dissemination of glial tumor cells in the brain parenchyma and their ability to modulate and control these cells make them premium potential targets [28]. The relationship between the influence of different chemokines on migration and dissemination of glial tumor cells is relatively well described in the literature [29,30]. Research involving chemokine and glial tumor cells has classically attempted to limit or suppress the phenomenon of chemoattraction inducing cell migration to decrease the distant spread of tumor cells [31,32,33,34,35,36,37,38]. However, so far, this approach has not been successfully deployed in the clinic in the treatment of primary brain tumors contrary to other types of cancer [39]. For example, cilengitide, an integrin inhibitor, emerged as a promising candidate in preclinical studies, but failed the test of randomized clinical studies [25]. The data presented here suggest a potential new avenue in the use of chemokines in the treatment of primary brain tumors. Unlike prior research approaches, we instead elected to use the strong attraction capacity of chemoattractant molecules to our advantage instead of trying to disrupt it. We investigated the concept of attracting and maintaining these cells in the vicinity of the tumor nucleus area, with the idea of using this approach as a therapeutic device. In this design, the chemoattraction would be used in a luring scheme to limit the spread of tumor cells and repatriate surrounding infiltrated cells to a limited area now containing a treatment modality. This device would be implanted in the tumor resection cavity at tumor relapse. Several research groups are working on the involvement of chemoattractants in the migration and dissemination of cancerous glial cells. However, these molecules seldom have an isolated effect on the migrationnal phenotype. Chemokines are classically cell signaling molecules which interact with a multitude of cellular components and are involved in various molecular mechanisms. This characteristic makes it difficult to associate a unique role to a specific chemokine [40]. Nonetheless, some chemokines express a dominant behavior. Based on a literature search, we identified three chemokines potentially involved in the migration of glial tumor cells whereas the effect on other phenotypic determinants such as proliferation were considered negligible [20,21,22]. In this study, CXCL10 and CCL2 chemokines displayed the most significant effects on the attraction and migration of glial tumor cells in vitro, as well as in vivo in the F98-Fischer rat model. Our results present compelling evidence that the chemotactic molecules studied play a major role in the attraction of neoplastic glial cells.

To evaluate the chemotactic effect of the selected chemoattractants in vitro, various methods were available. One of the most used is the “Boyden chamber” [41]. Nevertheless, we elected not to work with this method because this type of manipulation is classically used to assess the ability of cells to invade a medium, rather than to study cell migration per se. Hence, we instead decided to use the agarose drop assay as carried out by Ahmed’s group [27] and by Wiggins and collaborators [42] especially for its simplicity. Involved in several pathological processes such as autoimmune [43] and cancer diseases [44], CXCL10 is secreted by astrocytes in the central nervous system [45] and specifically activates CXCR3 [46]. In the present study, this chemokine was the best candidate to affect glial cell migration. Likewise, our in vitro experiments show the significant role of CXCL10 as a promoter of chemoattraction for F98 and U87MG glial tumor cells. CCL2 and its receptor CCR2, also lend interesting results. Some investigators reported that CCL2 has been primarily recognized in gliomas among the chemokine pathways involved in tumor associated macrophage (TAM) chemoattraction [47]. Here, we provide further evidence by in vitro assays that CCL2 is involved in the migration and invasion of glial tumor cells. CCL11 and its principal receptor CCR3, is another chemoattractant candidate that appears to be involved in the migration of glial tumor cells [48]. Nevertheless, we did not observe a significant effect of CCL11 in vitro on migration of F98 and U87MG cell lines. This can be explained by the possible interaction between CCL11 and the CCR2 receptor which is described by some researchers as being an antagonist of CCL11. Indeed, Ogilvie et al. have shown that the CCL2 acts as a competitive antagonist and inhibits the effect of CCL11 on chemotaxis and enzyme release [49]. This could also explain the poor chemotaxis effect observed by the combination of the three chemoattractants in our in vitro assays.

In vivo experiments performed with the F98 bearing rat model have also demonstrated the chemoattraction effect expressed by the selected chemokines. This syngeneic standardized model behaves in a predictable and reproducible fashion and adequately mimics the behavior of human malignant astrocytomas [50]. Using this model, significantly more peritumoral clusters were observed when CXCL10, CCL2 and CCL11 were implanted in the controlateral hemisphere. This suggest that the use of the GlioGel with chemokines significantly modified the migration behavior of F98 glial tumor cells. Interestingly, as a proof of principle, GlioGel with chemokines inoculation intratumorally produced an opposite effect: a significantly smaller number of peripheral clusters were counted. All these observations support our working hypothesis and the continued work on this concept.

For the IHC analysis, we used GFAP as a marker of astroglial injury and we denoted that the simple process of GlioGel inoculation increased staining in the non-tumor bearing hemisphere. The greatest marking was observed in the left hemisphere with the GlioGel contain CXCL10, suggesting activation of surrounding astrocytes [51]. Nestin is also used as a marker of migration phenotype and neural stem cells [52]. Labelling was profuse within the tumor in control condition and the inoculation of GlioGel with chemokines, especially with CXCL10, produced an increase in the marking. Even if we observed a slight increase in cell proliferation with the Ki-67 staining when GlioGel with chemokine was inoculated in the tumor, the same process induced a drastic fall in the number of peritumoral clusters, especially with CXCL10. This suggests that our approach induce a decrease in tumor cell dissemination in the brain parenchyma. Given that we intend to proceed to tumor resection in patients prior to the inoculation of the GlioGel containing chemoattractants, the impact on cell proliferation should be negligible. As only one slide per sample per group was surveyed, these results need to be considered with cautious. With the goal of investigating the risk of an inflammatory reaction, we proceeded to a panel of immunohistochemical labelling. These staining did not reveal a significative presence of inflammatory cell population. These observations comfort us in the clinical translation of this concept.

Finally, through this study, the ability of different chemotactic molecules, more specifically CXCL10 and CCL2, to attract and stimulate the migration of glial cancer cells in vitro and in vivo has been demonstrated. The release of chemoattractant impregnated GlioGel inoculated into the tumor cavity immediately after the tumor resection could then be used to attract the migrating tumor cells into or near the cavity. A localized treatment could then be applied to treat these cell remnants. This approach is presently used in the development of a new loco-regional multimodal treatment called GlioTrap. This promising approach consists of an implantable device made of GlioGel allowing release of chemokines and chemotherapeutic drugs in addition to being a container for high LET radioisotope.

## 4. Materials and Methods

### 4.1. Chemicals and Biologicals

Recombinant human CXCL10/IP-10 protein (catalog number 266-IP-010), CCL2/MCP-1 protein (catalog number 279-MC-010) and CCL11/Eotaxin protein (catalog number 320-EO-020) were purchased from R&D Systems (Oakville, ON, Canada). Synthetic silicate nanoplatelets (Laponite XLG) were obtained from Southern Clay Products, Inc. (Louisville, KY, USA).

### 4.2. Cell Culture

The murine F98 and human U87MG cell lines were obtained from American type culture collection (ATCC) and were grown in monolayer using a solution of Dulbecco’s modified eagle medium (DMEM) and Eagle’s minimal essential medium (EMEM) respectively. They were supplemented with 10% fetal bovine serum (FBS) and a mix of penicillin–streptomycin. In addition, U87MG was also supplemented by a mix of MEM Non-essential Amino-acid and Sodium phosphate. Cells were incubated at 37 °C in a humidified environment with 5% CO_2_ and propagated upon confluence, every 3 days for the F98 cells and every 4 days for the U87MG cells. All saline solution, medium and supplemented media were purchased from Wisent Bioproducts (St-Bruno, QC, Canada).

### 4.3. Migration Assay

A total of 0.1 gram (g) of low-melting point agarose (catalog number 15517-022) purchased from Intvitrogen (Carlsbad, CA, USA) was placed into a 100 mL beaker and diluted into 20 mL PBS to make a 0.5% agarose solution. This was heated on a hot plate in the cell culture hood until boiling, swirled to facilitate complete dissolution, and then taken from the heat. When the temperature cooled to 40 °C, 90 μL of agarose solution was pipetted into a 1.5 mL Eppendorf tube containing 10 μL of PBS only (control) or with chemoattractant (final solution at 100 nM). For most experiments 9 μL PBS plus 1 μL chemoattractant stock solution (10 µg/100 µL in PBS 0.1% with bovine serum albumin (BSA) = stock solution at 11.5 µM) was employed. In a 35 mm petri dish, two ten-microliter spots of agarose (one containing PBS only and the other instilled with one or more chemoattractant(s)) were pipetted as rapidly as possible and allowed to cool for ~2 h at 4 °C. Then, 1 mL of culture cells (F98 or U87MG) were added very slowly into spot-containing dishes to prevent detachment of the agarose spots. The 35 mm petri dishes were placed overnight in the incubator at 37 °C with 5% CO_2_ and analyzed by microscopy the next morning. Images were taken with the Infinity 2 Capture camera on Olympus CKX41 inverted microscope developed by Olympus Corporation (Shinjuku, TYO, Japan).

To perform the first analysis, evaluating the chemoattraction potential of the tested molecules, the surface of each image has been divided into a 4 tiers surface parcellation to measure distance of migration: zone 0 is inside of the agarose drop, and as we get farther from the drop, zone 1, zone 2 and zone 3 delimited by the orange box (Figure 1). Hence, zone 1 was delimited by the agglomeration of cells around the drop and zones 2 and 3 were arbitrarily separated to show areas where there was a visible discrepancy in cell density as we got further from the drop. The second type of analysis consisted in evaluating the migration distance of the cells inside the drop (zone 0). For all the experiments, the distances were measured between the wall of the agarose drop and the three most invasive cells identified at the tip of the migration fronts inside the drop, as represented by the double yellow arrow (Figure 1). The cells of each zone were counted using ImageJ software developed by National Institute of Mental Health (Bethesda, MD, USA). Data were reported in number of cells/mm^2^ for the chemoattraction potential and in mm for the migration distance inside the drop to allow an adequate comparison between the different conditions.

### 4.4. Animals

Adult male Fischer rats weighting 225 to 250 g were acquired from Charles-River laboratories (Montréal, QC, Canada). The animal care and experimentation were conducted following recommendations of the Canadian Council on Animal Care. The protocol was approved by the Committee on the Ethics of Animal Experiments of the Université de Sherbrooke (CIPA/CFPA-FMSS, Number # 465-18).

### 4.5. GlioGel Preparation

GlioGel was prepared according to our previously developed protocol [26]. In brief, a stock solution of 18% (*w*/*w*) gelatin type A obtained from porcine skin was prepared by dissolving gelatin in Milli-Q water and heating to 40 °C to ensure that the gelatin was completely dissolved. A 9% (*w*/*w*) synthetic silicate stock solution was made in 4 °C water to prevent gelation of laponite nanoplatelets and ensure complete dissolution of the nanoclay particles. Gelatin and silicate nanoplatelets were mixed at a 1:1 ratio by vortex with Milli-Q water at 3000 rpm for 5 min. This was followed by reheating the obtained gel and vortex for better dispersion of nanoplatelets. The shear-thinning gels were then stored at 4 °C. Chemokines were directly mixed into the GlioGel for a final concentration of 500 nM for in vivo experiments. Control GlioGel was prepared with an identical volume of PBS.

### 4.6. Glioma Implantation Model: Tumor Implantation and GlioGel Inoculation

F98 cells were suspended in non-supplemented warm MEM at a concentration of 2000 cells/µL. The implantation (10,000 cells in 5 µL) was performed as described by Blanchard et al. [53]. However, in this study, we modified the implantation coordinates as such: 1 mm anterior, 2 mm lateral to the bregma and 2.5 mm under the external table of the skull. These coordinates were chosen to allow the tumor cells to be more superficial, closer to the corpus callosum, that we wanted to use in this study as a preferential migration route. Indeed, several studies have shown that glioblastoma tumor cells use the white matter fibers or blood vessels to facilitate their migration [54]. Three days after glioma cells implantation, 10 μL of GlioGel with or without (control) chemoattractants was inoculated at the same coordinates as the implantation of the tumor cells. Two sets of experiments were performed: (1) the GlioGel was implanted either in intratumoral/ipsilateral (right hemisphere) or (2) into the mirror coordinates in the controlateral hemisphere to the tumor (left hemisphere). A total of 88 animals were accrued in these experiments.

### 4.7. Brain Processing

Fifteen days after glioma cells implantation, euthanasia was carried out by exsanguination and intracardiac perfusion of 60 mL formaldehyde 4% for histological analysis. Upon retrieval, brain specimens were fixed in a formalin solution for 48 h, cut in the coronal plane in 1 mm width slices using a dedicated brain matrix and embedded in paraffin. The blocks were sectioned at 3 µm intervals and the resulting slides were stained with H&E. All histological slices of brains were digitized with the Hamamatsu Nanozoomer in visible mode at 40× resolution and analyzed with the NDP.View 2 software developed by Hamamatsu Photonics K.K. (Hamamatsu city, Shizuoka Japan).

### 4.8. Immunohistochemistry Staining

GFAP, nestin, Ki-67, CD86, CD68 and Granulocytes were surveyed by immunohistochemistry. Brain sections were deparaffinized and processed for antigen retrieval by heat-induced sodium citrate buffer (10 mM, pH 6.0). Sections were then incubated with GFAP primary antibody diluted 1:200 (catalog number sc-33673) purchased from Santa Cruz Biotechnology (Santa Cruz, CA, USA), nestin primary antibody diluted 1:100 (catalog number sc-33677) purchased from Santa Cruz Biotechnology (Santa Cruz, CA, USA), Ki-67 primary antibody diluted 1:200 (catalog number ab16667) purchased form Abcam Inc. (Toronto, ON, Canada), CD86 (B7-2) primary antibody diluted 1:50 (catalog number sc-28347) purchased from Santa Cruz Biotechnology (Santa Cruz, CA, USA), CD68 primary antibody diluted 1:50 (catalog number sc-70761) purchased from Santa Cruz Biotechnology (Santa Cruz, CA, USA) or Granulocytes (HIS48) primary antibody diluted 1:50 (catalog number sc-19613) purchased from Santa Cruz Biotechnology (Santa Cruz, CA, USA) overnight at 4 °C. The secondary antibody used was coupled to a mouse or rabbit HRP depending on the primary antibody and was used directly without dilution (catalog number K400111-2) purchased from Agilent (Santa Clara, CA, USA). Hematoxylin counterstaining was then used to enhance the contrast. All histological slices of brains were digitized with the Hamamatsu Nanozoomer in visible mode at 40× resolution and analyzed with the NDP.View 2 software developed by Hamamatsu Photonics K.K. (Hamamatsu city, Shizuoka, Japan).

### 4.9. Immunohistochemical Quantification Image

Based on the prior work from our lab by Blanchette et al., who described a quantitative analysis technique based on indirect albumin immunohistochemistry to measure the extent of the BBB breach [55], we developed a similar method to detail our immunohistochemistry results quantitatively. We used ImageJ/Fiji software to perform different quantifications of the IHC images. This procedure was carried out in 3 main steps: (1) The outline of the immunohistochemical image was freehand selected and deleted, making it black and an image deconvolution was performed in H&E/DAB; (2) a threshold was then applied according to the marker to dichotomize the immunohistochemical labeling on each specimen. For GFAP, a threshold value <125 was used to consider labelling negative whereas a value > or equal to 125 was deemed positive. The threshold value for nestin was 145, and 195 for Ki-67; (3) the resulting images were then split along the midline to separate each hemisphere for the analysis. Then, each image was inverted to obtain IHC staining in white pixels. Finally, the sum of the points of each image was carried out (please see Appendix A for more details). These total pixel values, for each hemisphere were then reported in a graph to compare the different conditions (Figure 5).

### 4.10. Statistical Analysis

Data were analyzed by Shapiro–Wilk Normality tests, Fisher’s exact test and Student’s *t* test/Multiple *t* test (or non-parametric tests such as the Welch and Wilcoxon tests if necessary), to compare two treatments together. *p* values under 0.05 were considered statistically significant. All statistical analysis was done with GraphPad Prism 8 software (San Diego, CA, USA).

## Figures and Tables

**Figure 1 ijms-22-12150-f001:**
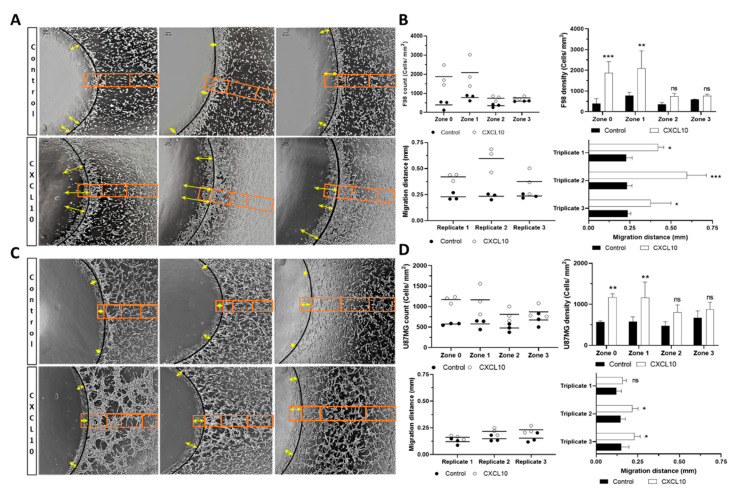
Measurement of chemotactic response for F98 and U87MG tumor cells. (**A**) Triplicate results of F98 cells obtained for chemotactic potency (orange box) and migration distance inside the agarose drop (yellow double arrow) in control condition or with CXCL10 in the environment. Cells located near the control drop which contains only PBS (Control) show a lower migration behavior than the cells in presence of the CXCL10 released by the agarose drop. (**B**) UPPER SEGMENT-Individual (*left*) and average (*right*) triplicate counts of cells in different zones of the control and the drop containing CXCL10 for F98 cells. LOWER SEGMENT-Individual (*left*) and average (*right*) triplicate value for migration distance inside the control and CXCL10 drop for the F98 cells. (**C**) Same as in (**A**) but for U87MG cells. (**D**) UPPER and LOWER SEGMENT-Same as in (**B**) but for the U87MG cells, *n* = 3 for each condition. *, *p* < 0.05, **, *p* < 0.01, ***, *p* < 0.001, ns, not significant. The asterisk is added above the value that is compared to the control on the left (**B**) or it is added next to the value that is compared to the control below (**D**).

**Figure 2 ijms-22-12150-f002:**
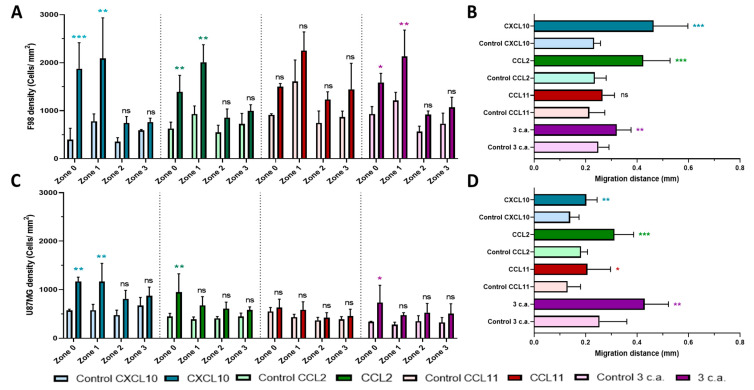
Summary of the Agarose drop assay for chemokines CXCL10, CCL2 and CCL11 individually and combination (3 c.a.). (**A**) Chemotactic potency of different chemokine and (**B**) average migration distance inside of the agarose drop for the F98 tumor cells. Same experiments in (**C**,**D**) for the U87MG cell line. Data obtained from images were taken 16 hours after initiating the test, *n* = 3 for each condition. *, *p* < 0.05, **, *p* < 0.01, ***, *p* < 0.001, ns, not significant. The asterisk is added above the value that is compared to the control on the left (**A**,**C**) or it is added next to the value that is compared to the control below (**B**,**D**).

**Figure 3 ijms-22-12150-f003:**
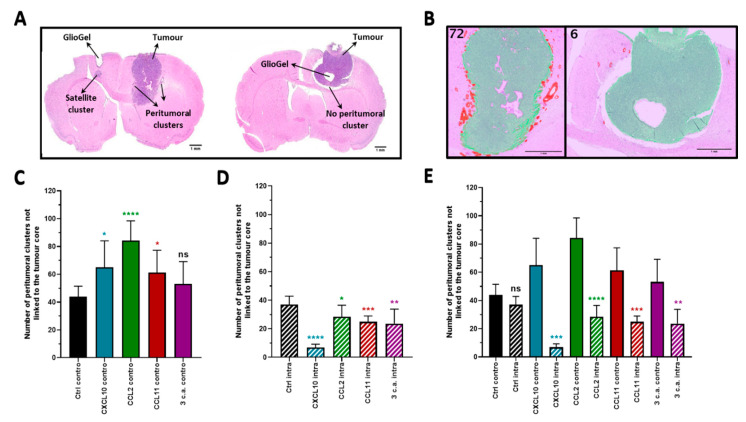
Brain histology of F98 glioma bearing rats for chemoattraction assays. Fischer rats were implanted with 10,000 F98 cells, GlioGel delivering chemoattractant were implanted 3 days later and brain collections were 12 days later (day 15 post tumor implantation) and processed for H&E coloration. (**A**) LEFT-GlioGel implanted into the controlateral hemisphere. The tumor has produced multiple peripheral clusters as well as satellite clusters in the controlateral hemisphere below the GlioGel inoculation point. RIGHT-GlioGel injected into the tumor (ipsilateral hemisphere). The tumor appears well restricted, showing almost no peripheral cluster. (**B**) Example of clusters count for (**A**) with the plugin Trainable Weka Segmentation in Fiji software. Red = clusters, Green = tumor core, Purple = brain background. (**C**–**E**) Glioma behavior according to the chemoattractant type and location. Contro = controlateral, Intra = intratumoral, Ctrl = Control, *n* = 7 to 10 animals depending on the group. *, *p* < 0.05, **, *p* < 0.01, ***, *p* < 0.001, ****, *p* < 0.0001, ns = not significant. The asterisk is added above the value compared to the control (**C**,**D**) or the corresponding “contro” group (**E**).

**Figure 4 ijms-22-12150-f004:**
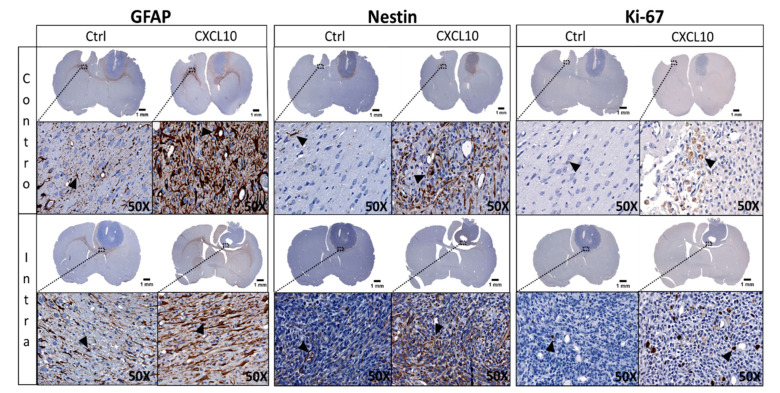
Brain immunohistology (GFAP, nestin and Ki-67) of F98 glioma bearing rats for the hemisphere having received GlioGel with or without chemoattractant. GlioGel implanted into the controlateral (Contro) or intratumoral (Intra) hemisphere without chemoattractant (Ctrl) or with CXCL10 chemoattractant. Immunohistochemical staining appear in brown (examples are pointed by the black arrowheads).

**Figure 5 ijms-22-12150-f005:**
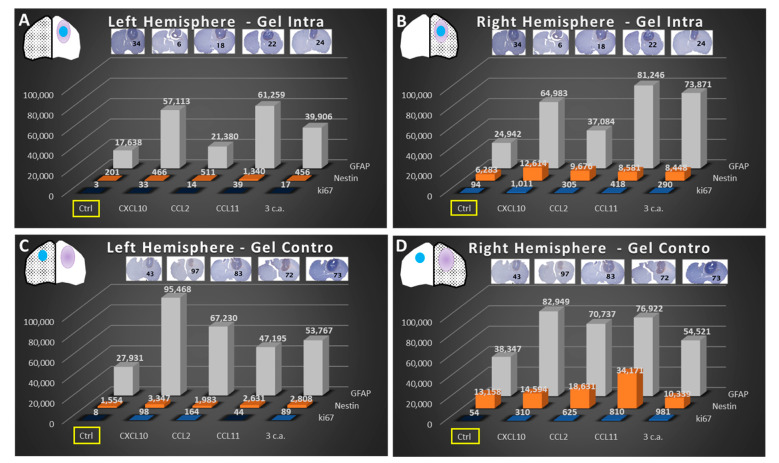
Quantification of the staining score of brain immunohistology of F98 glioma bearing rats for the hemisphere having received GlioGel with or without chemoattractant. (**A**) Fraction staining score for each marker and all chemokines of the ipsilateral hemisphere when the GlioGel is implant in the tumor. (**B**) Fraction staining score for each marker and all chemokines of the intratumoral hemisphere when the GlioGel is implant in the tumor. (**C**) Fraction staining score for each marker and all chemokines of the ipsilateral hemisphere when the GlioGel is implant in the ipsilateral hemisphere (without tumor). (**D**) Fraction staining score for each marker and all chemokines of the intratumoral hemisphere when the GlioGel is implant in the ipsilateral hemisphere (without tumor). Fraction staining score values for GFAP, nestin and Ki-67 appear in gray, orange and blue, respectively. Ctrl = Control. Black numbers in the right hemispheres represent the amount of satellite clusters. Blue circles represent the GlioGel and the texture in each hemisphere is for indicate which hemisphere is selected.

## Data Availability

Data will be provided if requested.

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
