# Peer review of "Chemoattraction of Neoplastic Glial Cells with CXCL10, CCL2 and CCL11 as a Paradigm for a Promising Therapeutic Approach for Primary Brain Tumors"

_ijms, 2021, doi:10.3390/ijms222212150_

Round 1

Reviewer 1 Report

Dery et al. investigated the role of chemoattraction in glioblastoma (GBM) invasion. They used hydrogel (GlioGel) loaded with CXCL10, CCL2 and CCL11 to test the effects of these chemoattractants in vitro in rat and human GBM cells and in vivo in a rat glioma model. The stated ultimate goal is to exploit the system for therapeutic purposes. They found that the chemokines induced tumor cell invasion in vitro. In vivo experiments demonstrated the capacity of specific chemoattractants to strongly attract neoplastic glioblastoma cells. The authors suggested that chemoattractant impregnated GlioGel inoculated into the tumor cavity immediately after the tumor resection could be used to attract the migrating tumor cells into or near the cavity for a more focused therapy.

This study presents an interesting experimental therapeutic strategy. The research is well performed and the data support the conclusions. The manuscript is well written.

The following minor issues should be addressed:

- As per the WHO, the designation “multiforme” is not used any more when referring to GBM. Please, refer to GBM as “glioblastoma” only.

- There is no convincing evidence that GBM originates in astrocytes. Astrocytomas are designated as such because they look like astrocytes and not because they originate from these cells.

- GBM does not make up 80% of all malignant brain tumors. Please, check CBTRUS and correct (the number is ~ 50% of all “primary” malignant brain tumors).

- Please, refrain from using evaluative words such as “unfortunately”. The text should describe the data in a neutral manner.

- The authors propose the use of chemoattractant impregnated GlioGel inoculated into the tumor cavity immediately after the tumor resection could then be used to attract the migrating tumor cells into or near the cavity. What would prevent the chemoattractant from diffusing into the brain parenchyma and causing more invasion? This issue merits some discussion.

Author Response

Response to the Editor

Editor, reviewers #1 and #2 have shown a great understanding of our manuscript and have made very pertinent comments and suggestions.

We responded to the editor's comments directly in the manuscript.

Here are our answers to the minor issues, as well as to the excellent comments and suggestions raised by reviewers.

Response to Reviewer 1 Comments

Point 1: As per the WHO, the designation “multiforme” is not used any more when referring to GBM. Please, refer to GBM as “glioblastoma” only.

Response 1: Updates according to the last WHO have been done. The designation “multiforme” was removed as proposed by reviewer #1. Corrections have been made at lines #15, #29, #34, and in the Abbreviations section.

Point 2: There is no convincing evidence that GBM originates in astrocytes. Astrocytomas are designated as such because they look like astrocytes and not because they originate from these cells.

Response 2: We acknowledge the fact that the cell of origin remains undetermined and could be neuronal stem cells. However, many authors consider the astrocyte as the cell of origin. (John Lin et al., 2017) Hence, to echo this concern of the reviewer, we have modified the statement at lines #35 to 37.

Point 3: GBM does not make up 80% of all malignant brain tumors. Please, check CBTRUS and correct (the number is ~ 50% of all “primary” malignant brain tumors).

Response 3: Reviewer #1 is right. The correction has been made at line #35.

Point 4: Please, refrain from using evaluative words such as “unfortunately”. The text should describe the data in a neutral manner.

Response 4: Reviewer # 1 mentions a good point. We have made the necessary changes at the line(s) #43, #208, #339, #342, and #381.

Point 5: The authors propose the use of chemoattractant impregnated GlioGel inoculated into the tumor cavity immediately after the tumor resection could then be used to attract the migrating tumor cells into or near the cavity. What would prevent the chemoattractant from diffusing into the brain parenchyma and causing more invasion? This issue merits some discussion.

Response 5: Reviewer #1 raises up an excellent question.  For now, as we have mentioned at line 117, we can only presume that the release of chemoattractant creates a gradient and that the migrating tumor cells follow this gradient toward the GlioGel. So far, we have always observed a unidirectional migration of cells to the source of chemoattractants (GlioGel) in all experimental conditions (in vitro and in vivo). We also know, from past experiments, that because of CSF circulation from the Virchow-Robins spaces, it is highly unlikely that chemoattractants emitted from the gel would penetrate significantly enough the brain parenchyma to induce another directional gradient then the one intended.

Ref: 1. Fortin D. Drug Delivery Technology to the CNS in the Treatment of Brain Tumors: The Sherbrooke Experience. Pharmaceutics. 2019;11(5):248. doi:10.3390/pharmaceutics11050248

Reviewer 2 Report

The authors presented a nice paper preliminarly exploring an interesting idea, i.e. to exploit the use of CXCL10, CCL2 and CCL11 to maintain neoplastic cells in the surgical cavity of an operated glioblastoma, riducing their infiltration, thus allowing local therapeutic strategies to be implemented. In their mind, this could possibly improve the effect of different local therapy by hitting more cells that are maintained closer to the surgical cavity. 

In order to preliminary evaluate the feasibility of such approach, they performed different ex-vivo and in-vivo experiments with 2 cell lines and in an orthotopic tumor rat model. In vitro assays resulted in an invasive cell rate 2-fold higher when chemokines were present in the environment (this effect was more pronounced with CXCL10). In vivo experiments demonstrated the capacity of these specific chemoattractants to strongly attract neoplastic glioblastoma cells, either when injected in the contralateral hemisphere or when injected in the tumoral area. 

I have two suggestions: 

1) the first two figures are somehow very similar and could maybe be condensed in one

2) it would have been good to explore the effect of tumor resection also in animal models, and evaluate the injection of GlioGel with and without chemoattractant directly in the surgical cavity after resection. This could confirm the authors' hypothesis that these chemoattractants are able to significantly reduce cell invasion at the surgical cavity, in order to replicate what they propose in humans as a subsequent step. 

Author Response

Response to the Editor

Editor, reviewers #1 and #2 have shown a great understanding of our manuscript and have made very pertinent comments and suggestions.

We responded to the editor's comments directly in the manuscript.

Here are our answers to the minor issues, as well as to the excellent comments and suggestions raised by reviewers.

Response to Reviewer 2 Comments

Point 1: The first two figures are somehow very similar and could maybe be condensed in one.

Response 1: Reviewer #2 is right in his comment. We had the same thought as reviewer #2 before submitting the manuscript.  However, we have decided to present our results in two figures to produce a more streamlined reading experience, to avoid overwhelming the reader with too much information into one figure.

If reviewer #2 agrees, we would like to keep these two figures separate for the mentioned reason.

Point 2: It would have been good to explore the effect of tumor resection also in animal models and evaluate the injection of GlioGel with and without chemoattractant directly in the surgical cavity after resection. This could confirm the authors' hypothesis that these chemoattractants are able to significantly reduce cell invasion at the surgical cavity, in order to replicate what they propose in humans as a subsequent step.

Response 2: This comment is in keeping with the ongoing experiments carried on in the lab. We are presently working on resection experiments and are in the process of completing and reporting such a standardized procedure in our animal model. As we did for the F98 Fischer rat characterization (Mathieu et al., 2007), in vivo BBBD manipulation (Blanchette et al., 2014), and in vivo irradiation experiments (Charest et al., 2013), we aim to develop a safe, robust, and reproducible resection method. Our objective is to perform resections for all the subsequent in vivo experiments on GBM, eventually including further chemoattraction experiments.

However, the resection method is not yet validated and cannot be confidently applied for the present study. We also firmly believe that the proof of principle of in vivo chemoattraction presented in this paper speaks for itself. Nonetheless, when the complete device will finally be tested, tumor resection will be performed prior to GlioGel insertion.

Round 2

Reviewer 2 Report

The authors have addressed the comments raised in the first revision. I still believe that the tumor resection model would be very useful to further confirm the authors' hypothesis.